# Uncertainty-aware Cycle Diffusion Model for Fair Glaucoma Diagnosis

**Ziheng Wang**[*1]                                              ZIHENG.WANG@KAUST.EDU.SA
[1] *Bioengineering, Biomedical Sciences Division, King Abdullah University of Science and Technology (KAUST), Thuwal, Saudi Arabia*

**Shuran Yang**[*2]                                                    SRYANG@XZMU.EDU.CN
[2] *School of Journalism and Communication, Xizang Minzu University, Xianyang, China*

**Yan Lin**[3]                                                          Y.LIN64@NCL.AC.UK
[3] *School of Computing, Newcastle University, Newcastle upon Tyne, UK*

**Wenrui Zang**[4]                                                    YC57670@UM.EDU.MO
[4] *Faculty of Health Sciences, University of Macau, Macao SAR, China*

**Yanda Meng**[†1]                                               YANDA.MENG@KAUST.EDU.SA

**Editors:** Accepted for publication at MIDL 2026

## Abstract

Fairness has become a critical ethical concern, particularly in AI-based healthcare applications. Data imbalance and limited sample size can lead to lower diagnostic performance. Consequently, this harms the fairness of AI when applied to real-world scenarios. Generative models, like diffusion models, offer a promising solution by generating diverse synthetic data to support underrepresented groups. This improves fairness and performance while mitigating privacy risks. We propose a shape-controlled framework that incorporates demographic information into an end-to-end diffusion model, along with an automatic selection strategy to identify overconfidently misclassified samples. These challenging samples are then augmented via the generative model to enhance its classification performance. The strategy also removes potentially misleading "lower-quality" synthetic samples. Two ophthalmic experts validated the clinical relevance and plausibility of our synthetic images through random external examination. Our method outperforms state-of-the-art methods on the Harvard-FairVLMed dataset in both fairness and diagnosis accuracy. Our code is available at https://github.com/WANG-ZIHENG/CCG.

**Keywords:** Fairness Learning, Image Synthesis, Diffusion Models, ControlNet

## 1. Introduction

Glaucoma is an irreversible optic nerve disorder that can lead to blindness if left untreated (Pereira et al., 2021). While AI-enhanced computer vision has been successfully applied to glaucoma diagnosis on retinal images (Sreng et al., 2020; Meng et al., 2022; Yu et al., 2025; Liu et al., 2025), these models often inherit demographic imbalances from their training data, resulting in systematically poorer performance for minority groups (Tian et al., 2023). A recent study (Luo et al., 2024a) reported that Black communities are more

---

[*] Contributed equally

[†] Corresponding author

than four times as likely to have undiagnosed glaucoma compared with white communities, highlighting the need to address fairness in AI-assisted glaucoma diagnosis.

Generative models have been explored in the medical domain (Kazerouni et al., 2023; Ktena et al., 2024), enriching training datasets to mitigate representation imbalance of certain populations or disease conditions. However, existing approaches (Yoon et al., 2023; Lyu and Wang, 2022) rarely consider fairness-related attributes during synthesis. Many prior methods (Li et al., 2024; Suresh et al., 2023) generate synthetic data based only on overall dataset distribution, which may yield redundant or less informative samples. Importantly, underrepresented subgroup samples often constitute many of the model's hard cases due to limited subgroup-specific training signals, as also observed in recent work (Ktena et al., 2024). In contrast, our method incorporates demographic identity information into the diffusion generation pipeline to support subgroup-aware generation, while introducing a behavior-driven selection mechanism that targets hard examples—i.e., samples misclassified with high confidence. By generating synthetic variants of these challenging cases, our approach improves both diagnostic performance and fairness.

We adopted the ControlNet-guided Stable Diffusion model (ControlNet-guided SD) (Zhang et al., 2023) to generate scanning laser ophthalmoscopy (SLO) images for glaucoma diagnosis. By integrating demographic identity information and clinical records as text prompts and using the optic disc segmentation mask as a shape control input, our model generates fairness-aware, shape-controlled synthetic images, enhancing diagnostic performance. Additionally, we design a Sorter module that leverages an overconfident error metric, calculated using prediction error and uncertainty, to automatically select challenging samples for data augmentation and subsequent training, while filtering out "lower-quality" synthetic images. Because the synthetic images are generated from overconfidently misclassified cases and are conditioned on consistent label and demographic attributes, the augmentation directly targets the classifier's failure modes. Guided by the Sorter rather than global data balancing, this targeted augmentation provides additional supervision exactly where the model underperforms, improving both diagnostic accuracy and subgroup fairness. Notably, among the saved synthetic samples, 100 of them were randomly selected and validated by ophthalmology experts for clinical relevance and plausibility, ensuring their applicability in real-world glaucoma diagnosis. In summary, our method improves glaucoma diagnosis and enhances model fairness across both majority and minority groups.

Our main contributions are: (1) We propose Cycle Control Generation (CCG), an end-to-end framework combining ControlNet-guided SD and a CNN classifier to generate clinically meaningful, demographically conditioned SLO images. (2) We introduce a behavior-driven sample selection strategy based on overconfident errors to identify high-confidence misclassifications. (3) We design a dynamic augmentation pipeline that iteratively updates challenging samples and removes misleading ones, guiding the model to focus on failure regions. (4) Our generated images show high visual quality and diagnostic relevance, validated through ophthalmologist evaluations and quantitative metrics.

## 2. Related Work

Diffusion models (Rombach et al., 2022; Kazerouni et al., 2023) have recently gained popularity for their ability to generate high-quality and diverse samples. For instance, diffusion

models with classifier guidance have been used to generate realistic and meaningful counterfactuals for retinal imaging (Ilanchezian et al., 2025). SynDiff (Özbey et al., 2023) outperforms existing methods in multi-contrast MRI and MRI-CT translation. ControlPolypNet (Sharma et al., 2024) synthesizes realistic colon polyp images from non-polyp frames, improving segmentation performance. UnIACorN (Niemeijer et al., 2025) leverages target-domain uncertainty and source-domain labels to synthesize labeled target-style OCT data, improving cross-domain segmentation. In this work, we similarly use diffusion-based synthesis, but incorporate fairness by using demographic identity information as text prompt inputs for the diffusion model.

Fairness in AI-based medical image analysis is an essential ethical issue. Recent studies in ophthalmic imaging have introduced demographic-aware datasets to address subgroup disparities. FairSeg (Tian et al., 2023) proposes an error bound scaling method that reweights the loss by group-specific error bounds, improving fairness. Similarly, FairDiff (Li et al., 2024) uses a two-stage diffusion framework that first generates cup-to-disc contours and then uses them as shape conditions in a ControlNet-guided SD to synthesize SLO images, aiming to balance subgroup representation and improve segmentation fairness. Unlike these methods, we adopt a behavior-driven strategy targeting overconfident misclassifications and use generative augmentation to address model weaknesses. This focus on challenging cases improves accuracy and fairness across subgroups.

## 3. Method

### 3.1. Metrics of Sample Selection Strategy

#### 3.1.1. Prediction Error

For a dataset $X = \{x_1, x_2, ..., x_n\}$ containing $n$ samples, the prediction error $e_1$ for a sample $x_1$ with label $g_1$ and model output logits $l_1$ is calculated as follows:

$$e_1 = |\sigma(l_1) - g_1|, \tag{1}$$

where $\sigma$ is the sigmoid function, the prediction error $e_1$ quantifies the absolute difference between the predicted probability $\sigma(l_1)$ and the ground truth label $g_1$. A large $e_1$ indicates that the model struggles with the sample, implying it is under-learned or difficult. Thus, $e_1$ serves as a useful signal for identifying samples needing more attention during training.

#### 3.1.2. Uncertainty

As emphasized in prior work (Ovadia et al., 2019), quantifying uncertainty is valuable for classification tasks. To leverage the value of uncertainty, we follow the method in (Meng et al., 2023) and add $T$ instances of random noise $\epsilon$ to the sample $x_1$, where $\epsilon \sim N(0, \sigma_{\text{noise}}^2)$ (with $\sigma_{\text{noise}}^2$ denoting the variance of the noise). This results in $\tilde{x}_{1,j} = x_1 + \epsilon_j$ for $j \in \{1, 2, ..., T\}$. The average output logits of the model $f$ for the sample $x_1$ are computed as $\bar{l}_1 = \frac{1}{T}\sum_{j=1}^{T} l_{1,j}$, where $l_{1,j}$ denotes the output of the model $f$ for $\tilde{x}_{1,j}$. Consequently, the uncertainty $u_1$ in the model's prediction for the sample $x_1$ is defined as follows:

$$u_1 = -\sum_k \sigma(\bar{l}_{1,k})log(\sigma(\bar{l}_{1,k})), \tag{2}$$

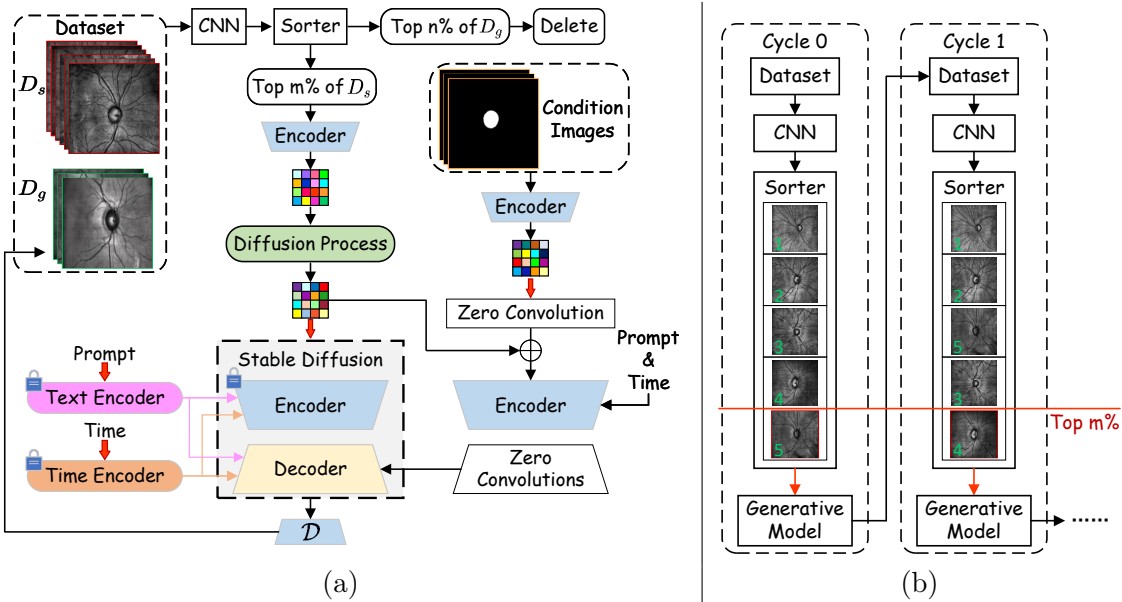

Figure 1: (a) Overview of the proposed Cycle Control Generation framework for generating synthetic SLO images. (b) Illustration of the dynamic update process of overconfidently misclassified samples automatically selected by the Sorter during training.

where $k$ represents the classes of the dataset and $\sigma$ is the sigmoid function. $u_1$ is used as an uncertainty measure derived from the model outputs under stochastic perturbations. The uncertainty measure $u_1$ reflects the model's predicted variability and confidence in its output for a given instance. A lower $u_1$ indicates greater stability and higher confidence, implying minimal influence from stochastic perturbations and noise on the prediction.

### 3.1.3. OVERCONFIDENT ERROR

Given that a smaller uncertainty $u$ indicates higher confidence, a large prediction error $e$ with small $u$ implies the model is confidently wrong—an *overconfident error*, which is especially harmful in medical diagnosis (Mukhoti et al., 2020; Lakshminarayanan et al., 2017). To quantify this, we define the *overconfident error value* $v_1$ for sample $x_1$ as the product of prediction error and confidence $(1 - u_1)$:

$$v_1 = e_1 \cdot (1 - u_1). \tag{3}$$

Here, $(1 - u_1)$ is a confidence factor, so samples with large prediction error and low uncertainty receive higher scores. We then use $v_1$ to identify overconfidently misclassified samples and perform targeted augmentation by generating label-consistent synthetic variants, where uncertainty helps prioritize failure cases over sample underrepresentation alone.

### 3.2. Cycle Control Generation

As illustrated in Fig. 1(a), the proposed Cycle Control Generation (CCG) is an end-to-end framework that integrates a CNN-based classification model, EfficientNet (Tan and Le, 2019), with a ControlNet-guided SD (Zhang et al., 2023). We use ControlNet to condition synthesis on optic disc segmentation masks and on demographic and clinical text prompts, so the generator produces shape-controlled and subgroup-aware SLO images that align with downstream diagnosis. The pipeline is organized around the overconfident error in Eq. 3, which scores each sample and drives targeted augmentation and pruning during training.

Concretely, the CNN outputs evaluation scores $v$ for the source dataset $D_s$ according to Eq. 3. The Sorter module ranks the samples based on their $v$ values and selects the top $m\%$ samples, which are identified as overconfidently misclassified by the CNN model. These challenging samples, corresponding text prompts (including demographic identity information and clinical records), and segmentation masks are fed into the ControlNet-guided SD (Zhang et al., 2023) to generate synthetic samples. The generated images are then stored in a generated dataset $D_g$, thereby expanding the number of challenging samples and improving classification performance in the subsequent training cycle.

Starting from Cycle 1, the CNN model is fine-tuned using both $D_s$ and $D_g$, initializing from the model weights that achieved the highest AUC on the validation set in the previous cycle. The CNN outputs $v$ values for both $D_s$ and $D_g$. The top $m\%$ samples from $D_s$, based on their $v$ values, are selected for the current cycle's generation process. Our method then ranks all samples in $D_g$ according to their $v$ values and removes the top $n\%$ samples. Here, "lower-quality" synthetic samples in $D_g$ should not be confused with the challenging source samples selected from $D_s$. In our framework, high-$v$ samples in $D_s$ are informative hard cases used to guide generation, whereas high-$v$ samples in $D_g$ are treated as less reliable synthetic training samples and are therefore removed. This prevents the model from being misled in subsequent training iterations.

Fig. 1(b) illustrates the dynamic update process of overconfidently misclassified samples automatically selected by the Sorter module during training. At Cycle 0, Sample 5 is identified as an overconfident error and is used for sample generation. After its sample count is augmented, the CNN may learn to classify it better in Cycle 1, resulting in a lower $v$ value. Consequently, it is no longer selected by the Sorter module. Instead, Sample 4, now identified as an overconfident error, is selected for generation. As training progresses, the set of overconfidently misclassified samples changes dynamically. In other words, the model focuses on refining challenging samples, leading to improved classification performance.

## 4. Experiments

### 4.1. Datasets

We utilized a subset of 7,363 images from the Harvard-FairVLMed (FairVLMed) dataset (Luo et al., 2024a), as used in (Wang et al., 2025), including 5,266 images for training, 692 images for validation, and 1,405 images for testing. Each SLO fundus image is accompanied by demographic identity group information and clinical records written by ophthalmologists. During training, synthetic SLO images were iteratively generated and incorporated into the training set. By the end of training, 510 synthetic images had been added.

| Attribute | Group | train | val | test | % | generated |
|-----------|-------|-------|-----|------|-----|-----------|
| Glaucoma | With | 2414 | 304 | 638 | 45.58 | 310 |
| | Without | 2852 | 388 | 767 | 54.42 | 200 |
| Race | Asian | 396 | 41 | 98 | 7.27 | 43 |
| | Black | 651 | 85 | 177 | 12.40 | 77 |
| | White | 4219 | 566 | 1130 | 80.33 | 390 |
| Gender | Female | 3055 | 406 | 789 | 57.72 | 224 |
| | Male | 2211 | 286 | 616 | 42.28 | 286 |
| Ethnicity | Non-Hispanic | 4789 | 626 | 1271 | 90.81 | 466 |
| | Hispanic | 477 | 66 | 134 | 9.19 | 44 |

Table 1: Attribute and group distribution in original and generated datasets.

A detailed breakdown of the dataset's attributes, including race, gender, and ethnicity distribution across the training, validation, test sets, and generated synthetic SLO images, is provided in Table 1. The percentage (%) column in the table represents the proportion of each attribute across the entire dataset (training, validation, and test sets combined). For example, 45.58% indicates that samples with glaucoma make up 45.58% of the total dataset. We also provide the data distribution across various attributes and groups for the generated dataset, shown in the last column in Table 1.

## 4.2. Text and Shape Conditioning

We condition synthesis on both text and shape. For text, we convert demographic and clinical attributes from FairVLMed (Luo et al., 2024a) into short prompts (e.g., "Male, White, Non-Hispanic, with Glaucoma"). This encourages the generated images to better reflect diverse subgroups and facilitates subgroup-aware fairness evaluation. Diagnosis information is used solely for generation and no parameters are shared with the classifier, preventing information leakage. For shape, we fine-tuned a pretrained TransUNet (Chen et al., 2021) on Harvard FairSeg (Tian et al., 2023) and used it to segment optic disc regions in FairVLMed (Luo et al., 2024a). The resulting masks serve as control inputs to the generator.

## 4.3. Implementation Details and Evaluation Metrics

We run our CCG framework for 5 cycles in PyTorch on an NVIDIA RTX 4090. Each cycle consists of: (1) fine-tuning the CNN classifier on the current dataset $D_s \cup D_g$ for 30 epochs; (2) scoring and ranking samples with the Sorter using the overconfident error $v$; (3) removing the top $n\%$ synthetic samples in $D_g$ and selecting the top $m\%$ hard cases from $D_s$, both ranked by $v$, to fine-tune ControlNet for 2 epochs and generate synthetic images; and (4) augmenting $D_g$ with the newly generated samples for the next cycle. In all experiments, we set $n = 50$ and $m = 5$. $n = 50$ was selected from candidate pruning ratios (e.g., 40%, 50%, and 60%) based on validation AUC and subgroup performance, while $m = 5$ controls the proportion of hard cases used to guide generation. We use AdamW with learning rate 1e−5, weight decay 6e−5, batch size 16, and binary cross-entropy loss for classification. We set

$T = 8$ in Eq. 2, following prior work (Meng et al., 2023). All remaining hyperparameters are chosen by cross-validation on the training set. To ensure a fair comparison, all competing methods are trained with the same total number of CNN fine-tuning epochs as our method.

We evaluate classification with ACC, AUC, Precision (Prec.), Sensitivity (Sens.), and F1. Fairness is assessed with Demographic Parity Difference (DPD) (Agarwal et al., 2018, 2019) and Difference in Equalized Odds (DEOdds) (Agarwal et al., 2018). Image quality is measured with Fréchet Inception Distance (FID) (Heusel et al., 2017), Learned Perceptual Image Patch Similarity (LPIPS) (Zhang et al., 2018), Structural Similarity (SSIM) (Wang et al., 2004), and Multiscale-SSIM (MS-SSIM) (Wang et al., 2003), complemented by expert review. Classification metrics are reported as percentages, and image quality metrics are reported as raw scores.

## 5. Results

### 5.1. Classification Performance and Fairness

The *Baseline*, as shown in Tables 2 and 3, is the framework that ablates the Sorter module and generative model, meaning without generative operations and uses only the classification model to classify the source data. The compared fairness methods include FairCLIP (Luo et al., 2024a), FIN (Luo et al., 2024b), FairDomain (Tian et al., 2024), and FairVision (Luo et al., 2023), which improve fairness through Sinkhorn-based distribution alignment, fair identity normalization, fair identity attention under domain shift, and fair identity scaling, respectively. Unlike these methods, our CCG framework introduces behavior-driven hard-case selection, cyclic refinement, and shape-controlled generative augmentation. Table 2 shows that our method consistently outperforms previous related methods across all performance metrics on the FairVLMed dataset, significantly surpassing the baseline with notable improvements. Specifically, our method shows a 6.78% increase in ACC, 8.47% in AUC, 12.27% in Precision, 5.27% in Sensitivity, and an 8.11% gain in F1-score compared to the baseline. Notably, the simultaneous improvement in both precision and sensitivity in AI-assisted medical diagnosis indicates a reduction in both false negatives and false positives. This enhancement strengthens disease detection capabilities in medical image analysis, which is crucial for early diagnosis.

| Method | ACC $\uparrow$ | AUC $\uparrow$ | Prec. $\uparrow$ | Sens. $\uparrow$ | F1 $\uparrow$ |
|---|---|---|---|---|---|
| Baseline | 69.47 | 75.93 | 71.46 | 60.41 | 65.50 |
| FairCLIP (Luo et al., 2024a) | 74.03 | 82.78 | 83.41 | 61.25 | 70.63 |
| FIN (Luo et al., 2024b) | 74.86 | 80.43 | 75.86 | 65.49 | 70.29 |
| FairDomain (Tian et al., 2024) | 73.91 | 80.16 | 81.47 | 62.83 | 70.95 |
| FairVision (Luo et al., 2023) | 74.78 | 81.29 | 75.55 | 65.41 | 70.12 |
| Our work | **76.25** | **84.40** | **83.73** | **65.68** | **73.61** |

Table 2: Testing the effectiveness of our work, we compared its overall performance with other methods. The best performance results are highlighted in bold.

Table 3 presents the classification performance and fairness metrics of our method across various demographic attributes. In terms of fairness, our approach consistently outperforms existing methods by achieving the lowest DPD and DEOdds values across all attributes, indicating better mitigation of demographic disparities. For classification performance, our method achieves balanced improvements across groups regardless of their proportion in the dataset (as shown in Table 1), without compromising any individual group's performance. For the Ethnicity attribute, both the non-Hispanic group (90.81%) and the Hispanic group (9.19%) achieved at least a 2% AUC improvement over other methods. A similar trend of balanced improvement was observed for the Race and Gender attributes.

| Attribute | Method | DPD ↓ | DEOdds ↓ | Group-wise AUC ↑ | | |
|---|---|---|---|---|---|---|
| | | | | Asian | Black | White |
| Race | Baseline | 14.90 | 14.38 | 75.53 | 72.26 | 76.46 |
| | FairCLIP | 14.19 | 9.72 | 84.92 | 78.29 | 83.14 |
| | FIN | 7.79 | 13.45 | 85.78 | 79.04 | 80.18 |
| | FairDomain | 7.94 | 3.51 | 80.75 | 77.91 | 80.63 |
| | FairVision | 14.73 | 9.72 | 86.41 | 77.30 | 81.26 |
| | Our work | **6.25** | **3.29** | **87.03** | **79.98** | **84.29** |
| | | | | Female | Male | |
| Gender | Baseline | 7.69 | 13.92 | 74.51 | 77.78 | |
| | FairCLIP | 2.17 | 5.94 | 80.82 | 84.90 | |
| | FIN | 1.91 | 5.91 | 78.64 | 82.41 | |
| | FairDomain | 5.22 | 9.96 | 77.08 | 83.79 | |
| | FairVision | 2.16 | 4.63 | 79.76 | 82.73 | |
| | Our work | **1.74** | **4.38** | **82.81** | **86.09** | |
| | | | | Non-Hispanic | Hispanic | |
| Ethnicity | Baseline | 20.08 | 24.69 | 75.66 | 73.18 | |
| | FairCLIP | 2.71 | 3.94 | 82.19 | 80.03 | |
| | FIN | 18.46 | 23.88 | 79.95 | 78.41 | |
| | FairDomain | 12.53 | 13.45 | 79.72 | 77.65 | |
| | FairVision | 14.29 | 19.48 | 80.87 | 78.39 | |
| | Our work | **2.64** | **3.53** | **84.28** | **82.30** | |

Table 3: Fairness metrics and group-wise AUC across demographics, comparing our method, baseline, and prior work (FairCLIP (Luo et al., 2024a), FIN (Luo et al., 2024b), FairDomain (Tian et al., 2024), FairVision (Luo et al., 2023)).

It is important to note that demographic sample size does not necessarily determine classification performance. For instance, although the Female group constitutes 57.72% of the dataset—more than the Male group at 42.28%—all methods still show lower performance on the Female subgroup (Table 3). This highlights that fairness disparities are not caused solely by demographic imbalance, and that simply increasing samples or improving average accuracy does not ensure fair learning across attributes (Deho et al., 2023). By incorporating uncertainty into the overconfident error, our method prioritizes confidently wrong samples, enabling targeted augmentation of subgroup-specific failure cases rather than uniform sam-

ple expansion. Moreover, fairness-driven objectives, such as loss terms based on inter-group differences, may fail to capture subgroup-specific failure mechanisms and sometimes reduce disparities by lowering performance on stronger groups. In contrast, our method does not explicitly rebalance data according to demographic labels, but instead identifies overconfidently misclassified samples and applies semantic-preserving augmentation with a generative model. This behavior-driven strategy enables the model to learn from failure regions and yields consistent performance gains across subgroups, thereby improving fairness.

## 5.2. Ablation Experiment

### 5.2.1. Effect of Sample Selection Metrics

As shown in Section 5.1, our framework improves both classification and fairness over the Baseline. To assess the role of Overconfident Error (Eq. 3), we compared it with Prediction Error (Eq. 1) and Uncertainty (Eq. 2) individually. Results in Table 4 indicate that Overconfident Error yields superior performance across all five metrics, with gains of at least 3% in ACC, Precision, Sensitivity, and F1-score, and 5% in AUC. This confirms it as a more effective criterion for selecting challenging samples and filtering lower-quality synthetic ones.

| Method | ACC ↑ | AUC ↑ | Prec. ↑ | Sens. ↑ | F1 ↑ |
|---|---|---|---|---|---|
| Prediction Error | 71.32 | 78.79 | 78.24 | 61.12 | 68.63 |
| Uncertainty | 73.14 | 79.26 | 80.51 | 62.57 | 70.42 |
| Our work | **76.25** | **84.40** | **83.73** | **65.68** | **73.61** |

Table 4: Performance comparison of different sample selection strategies.

### 5.2.2. Contribution of the Cyclic Framework

To evaluate the effectiveness of our approach, we generated a comparable number of synthetic images using ControlNet, matching the total number of saved images in $D_g$. These images were directly added to the training set, referred to as CNN + Synth in Table 5. The results in Table 5 show that, although CNN + Synth also improves model classification performance compared to training on the source dataset with CNN alone, our method outperforms CNN + Synth across all metrics. Specifically, our approach surpasses CNN + Synth by 3.06% in ACC, 2.88% in AUC, 4.06% in Precision, 1.76% in Sensitivity, and 2.68% in F1-score.

| Method | ACC ↑ | AUC ↑ | Prec. ↑ | Sens. ↑ | F1 ↑ |
|---|---|---|---|---|---|
| CNN | 69.47 | 75.93 | 71.46 | 60.41 | 65.50 |
| CNN + Synth | 73.19 | 81.52 | 79.67 | 63.92 | 70.93 |
| CNN + 2Synth | 73.02 | 81.86 | 77.91 | 64.50 | 70.57 |
| CNN + 5Synth | 73.16 | 81.57 | 76.43 | 64.86 | 70.07 |
| Our work | **76.25** | **84.40** | **83.73** | **65.68** | **73.61** |

Table 5: Ablation study comparing direct synthetic augmentation with our work.

We further report CNN + 2Synth and CNN + 5Synth in Table 5, where "2Synth" and "5Synth" denote synthetic datasets with twice and five times the size of $D_g$. Results show

that simply enlarging synthetic data does not improve performance, as many generated samples are easily classified and add limited information, potentially leading to overfitting. In contrast, our method augments difficult samples, enabling the classifier to learn from challenging cases and achieve better generalization.

We also evaluated a variant without synthetic image filtering and observed an approximately 1% drop in ACC, AUC, and F1, indicating that retaining all generated samples, including lower-quality ones, can negatively affect downstream training, whereas pruning improves cyclic augmentation. This aligns with prior studies that apply selective augmentation or quality control before downstream use (Xue et al., 2021; Sharma et al., 2024). A complementary direction would be to further disentangle the individual contribution of demographic identity prompts within the generative conditioning. As demographic prompts in our framework also serve as the attribute assignment mechanism for synthetic samples, exploring alternative protocols for isolating their effect is an interesting avenue that we leave for future work.

### 5.3. Evaluation of Generated Data

#### 5.3.1. Qualitative Quality

To validate clinical quality, two experienced ophthalmologists independently evaluated 100 randomly sampled synthetic SLO images from $D_g$, without diagnostic prompts or labels to avoid bias. Each image was judged as "suggestive of glaucoma" or "not suggestive," primarily based on the cup-to-disc ratio (Haider et al., 2023). Experts noted that the images showed clinically relevant features, including clear optic disc boundaries, accurate cup-to-disc visualization, minimal artifacts, and physiologically consistent vascular structures, contributing to high confidence in their evaluations. The intraclass correlation coefficient (ICC) (Koo and Li, 2016) exceeded 0.95, indicating excellent interrater agreement. The evaluation achieved nearly 90% diagnostic accuracy compared to the intended generation labels, and as shown in Fig. 2, expert assessments of "with Glaucoma" and "without Glaucoma" closely matched the generation conditions, confirming the clinical validity of the synthetic images.

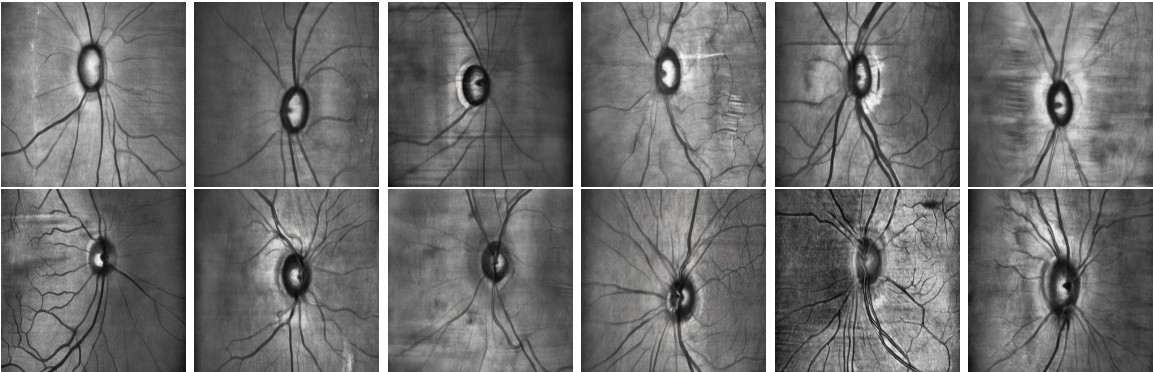

Figure 2: Synthetic SLO images generated with text prompts: first row "with Glaucoma," second row "without Glaucoma."

### 5.3.2. QUANTITATIVE QUALITY

We compute the FID, SSIM, MS-SSIM, and LPIPS metrics for the generated images, comparing them with the source dataset under the "With Glaucoma" and "Without Glaucoma" categories. As shown in Table 6, our method consistently outperforms ControlNet across all metrics. These results demonstrate the effectiveness of our approach in producing clinically meaningful, high-quality images, which also partially supports the conclusions drawn from the expert qualitative assessment. Our work is trained on challenging samples that may include complex or uncommon pathological features and variations in structure, angle, and clarity, enabling the model to learn richer details.

| Metric | Method | With Glaucoma | Without Glaucoma |
|---|---|:---:|:---:|
| FID ↓ | ControlNet | 56.50 | 98.20 |
| | Our work | **43.40** | **63.50** |
| LPIPS ↓ | ControlNet | 0.5187 | 0.5365 |
| | Our work | **0.4990** | **0.4879** |
| SSIM ↑ | ControlNet | 0.3136 | 0.3301 |
| | Our work | **0.3411** | **0.3848** |
| MS-SSIM ↑ | ControlNet | 0.2402 | 0.2278 |
| | Our work | **0.2459** | **0.2417** |

Table 6: Quantitative evaluation of the synthetic image quality generated by our proposed method compared to ControlNet (Zhang et al., 2023) generation.

## 6. Conclusion

In this work, we propose Cycle Control Generation, an end-to-end framework that integrates a CNN classifier with a ControlNet-guided diffusion model to generate clinically relevant synthetic SLO images. By leveraging a behavior-driven Sorter module to identify and augment overconfidently misclassified samples while filtering out low-quality generated ones, our method improves diagnostic accuracy and fairness. Extensive experiments on the FairVLMed dataset demonstrate superior performance over advanced methods. While our framework uses demographic information to enhance fairness, it also applies to datasets without such attributes by improving the classification of challenging samples. Within our framework, demographic information is incorporated as part of the generative conditioning to support subgroup-aware synthesis, and further disentangling its individual contribution within the generative pipeline is a natural extension that we leave for future work.

## Acknowledgments

This work is supported by YM's KAUST baseline research funding BAS/1/1121-01-01.

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
