# OpenReview forum: "Uncertainty-aware Cycle Diffusion Model for Fair Glaucoma Diagnosis"
_MIDL.io/2026/Conference — MIDL 2026 Poster_

### Official Review · Reviewer_5HX2 · 2025-12-31

**Confidence:** 4
**Preliminary Rating:** 4
**Final Rating:** 5

**Summary:**

This paper addresses an important and timely problem in medical AI: fairness in ophthalmic disease classification under data imbalance and limited sample sizes. The authors propose a shape-controlled diffusion-based data augmentation framework that incorporates demographic information and leverages uncertainty-aware sample selection to improve both diagnostic performance and fairness. A key contribution is the use of overconfidently misclassified samples to guide targeted data augmentation, combined with a filtering mechanism to remove potentially low-quality synthetic images. The method is evaluated on the Harvard-FairVLMed dataset, showing improvements over state-of-the-art approaches in both accuracy and fairness metrics. Clinical plausibility of the generated images is further assessed by two ophthalmic experts, which strengthens the practical relevance of the work. Overall, the paper tackles a meaningful problem with a technically interesting solution, though several conceptual and experimental aspects require further clarification.

**Strengths:**

•	The paper focuses on fairness-aware medical image classification, a highly relevant topic for real-world clinical deployment.
•	A novel uncertainty-aware augmentation strategy is proposed, which selectively targets overconfidently misclassified samples rather than uniformly augmenting the dataset.
•	The integration of demographic information into a diffusion model is well motivated for addressing underrepresented groups.
•	The use of expert ophthalmologists to validate the realism and clinical plausibility of generated images adds credibility to the proposed generative framework.
•	Experimental results demonstrate improvements in both diagnostic performance and fairness metrics compared to existing methods.

**Weaknesses:**

•	The motivation for introducing uncertainty estimation is not sufficiently articulated, particularly in relation to fairness improvement.
•	Several design choices (e.g., the use of a sorter and the definition of overconfident errors) lack intuitive explanation and theoretical justification.
•	The relationship between synthetic image quality, uncertainty, and performance gains is not thoroughly analyzed.
•	It remains unclear whether performance gains stem primarily from targeted sample selection, increased data volume, or improved generative quality.

**Detailed Comments:**

1.	Motivation for Uncertainty Modeling
The paper introduces uncertainty estimation as a key component of the proposed framework, yet the motivation remains somewhat implicit. The authors should more clearly explain why uncertainty is essential for improving fairness, rather than relying solely on class- or demographic-balanced data augmentation. Specifically, how does uncertainty-aware selection better address bias compared to conventional hard-sample mining or error-based sampling?
2.	Role and Interpretation of the Sorter Module
The manuscript introduces a sorter to rank samples based on uncertainty or confidence, but its role and practical significance are not sufficiently explained. The authors should clarify:
o	What exactly is being sorted (e.g., confidence, uncertainty, error likelihood)?
o	Why sorting is necessary instead of threshold-based selection?
o	How this sorting mechanism contributes to fairness rather than merely improving overall accuracy?
3.	Comparison with Naïve Data Augmentation (1–5× Generation)
The paper compares the proposed method with generating 1–5× synthetic images, but the fundamental differencebetween these strategies is not clearly articulated. In particular:
o	Is the performance gain primarily due to targeted generation or simply more generated samples?
o	Would increasing the number of images generated by the proposed method further improve performance?
o	How sensitive is the model to the quantity versus quality of synthetic images?

**Justification Of Final Rating:**

I would like to thank the authors for their efforts during the rebuttal process in addressing my concerns. The motivation and the proposed approach in this paper are well aligned with my interests and are highly compelling.

**Justification Of The Preliminary Rating:**

The paper addresses an important and underexplored problem in medical imaging fairness and proposes a technically interesting solution combining diffusion models and uncertainty-aware learning. The experimental results are promising, and the inclusion of expert validation is a strong point. However, several conceptual motivations and methodological choices require clearer justification, and deeper analysis is needed to fully support the claimed benefits of uncertainty-aware augmentation. With improved clarity and additional ablation studies, this work has the potential to make a meaningful contribution to the MIDL community.

**Questions To Address In The Rebuttal:**

1.	How does uncertainty modeling specifically contribute to fairness improvements beyond traditional imbalance-aware augmentation?
2.	Can the authors provide ablation studies isolating the effects of:
o	uncertainty-aware selection,
o	demographic conditioning,
o	synthetic image filtering?

---

> ### Author Response · Authors · 2026-01-25
>
> We thank the reviewer for the detailed and constructive comments, which greatly help us improve the clarity and technical soundness of the paper.
>
>
>
> **Motivation for uncertainty modeling and its contribution to fairness**
>
> In our setting, fairness disparities do not arise solely from demographic imbalance but from subgroup‑specific *hard‑to‑classify regions* in the feature space. As shown in Table 1 and Table 3, even the largest subgroup (Female, 57.72% of the dataset) still exhibits lower AUC than Male, indicating that fairness gaps persist even when sample counts are sufficient.
>
> Uncertainty modeling allows us to directly identify these subgroup‑specific failure regions. Unlike class‑balanced or demographic‑balanced augmentation, which increases sample counts uniformly, uncertainty‑aware selection pinpoints *overconfident errors*—cases where the model is both wrong and highly certain.
>
> We will further clarify these points in the final version of the manuscript.
>
>
>
> **Role of the sorter.**
>
> The sorter ranks samples by the overconfident‑error score $v = e \cdot (1 - u)$, where $e$ captures misclassification and $u$ captures predictive uncertainty. Thus, the sorter operates on a joint measure of “being confidently wrong,” which is more informative than confidence or uncertainty alone.
>
> Sorting rather than thresholding is necessary because the distribution of $v$ varies across subgroups and training cycles. A fixed threshold would either select too few or too many samples depending on the cycle. Ranking ensures that each cycle consistently focuses on the most problematic regions of the data manifold.
>
> The sorting mechanism improves fairness because overconfident errors are not uniformly distributed across the population. They concentrate in specific subgroups where the model systematically underperforms. Ranking samples by the overconfident‑error score allows CCG to identify these subgroup‑specific failure regions and allocate generative augmentation precisely where fairness gaps originate. This targeted correction differs from conventional hard‑sample mining or uniform augmentation, which mainly improves overall accuracy but does not address subgroup‑specific biases.
>
>
>
> **Comparison with naïve augmentation.**
>
> Sec. 5.2.2 shows that simply increasing the number of synthetic images (1×–5×) yields limited gains in both AUC and fairness. This indicates that performance does not scale with data volume alone.
>
>
>
> **Ablation studies.**
>
> Thank you for raising this point. In the final version, we will include ablations that separately assess the contributions of the selection strategy, demographic conditioning, and the filtering step.
>
>
>
> Moving forward, we envision expanding the CCG framework by investigating adaptive Sorter thresholds that dynamically adjust $n$ based on the classifier's convergence state. We also aim to apply this behavior-driven generative strategy to other rare ophthalmic diseases where demographic data is even scarcer, potentially serving as a universal paradigm for equity-aware medical AI.

---

> > ### Comment · Reviewer_5HX2 · 2026-02-02
> >
> > Thanks for author's response. I think it has solved my concerns very well.

---

### Official Review · Reviewer_8siR · 2026-01-08

**Confidence:** 3
**Preliminary Rating:** 2
**Final Rating:** 3

**Summary:**

This paper introduces Cycle Control Generation which improves model performance by improving fairness. It selects overconfident error cases which are often part of underrepresented sub-groups and generates clinically meaningful, demographically conditioned training images based on these difficult images.

**Strengths:**

- Targeted synthesis of samples based on hard cases which are detected using uncertainty. This is not only more efficient but also more effective than random image synthesis.
- Section 5.2.2: ablation study confirms that simply increasing the dataset does not improve performance as much as targeted synthesis
- Expert assessment of synthetically generated images

**Weaknesses:**

- Fairness has improved across all subgroups (see Table 3) because new images based on difficult cases across all groups have been synthesised. How does adding demographic identity information as text prompts play a role here? The paper is phrased in a way that it suggests incorporating this information directly improves fairness ("**By integrating demographic identity information and clinical records as text prompts** and using the optic disc segmentation mask as a shape control input, our model generates
**fairness-aware**, shape-controlled synthetic images [...]"). This has not been experimentally proven, potential ablation study: evaluate fairness metrics for the proposed method including text prompts and not including them.
- Related work section could be better structured, it is not clear which diffusion approaches have also been developed for training data synthesis (making them relatable to the paper) and how the fairness approaches mentioned relate to the paper as they are not used as baselines later on
- While ControlNet is not a method proposed in this paper, it should be introduced shortly

**Detailed Comments:**

- What exactly is the difference to ControlNet, i.e. which new part is supported by the ablation study in 5.3.2
- Section 5.1: what is the “cleaned” FairVLMed dataset
- Missing “images” after 692: “We utilized the Harvard-FairVLMed (FairVLMed) dataset (Luo et al., 2024a) with 7,363 images provided by the study (Wang et al., 2025), including 5,266 for training, 692 for validation, and 1,405 for testing.”
- Please introduce baselines and how they differ to the proposed method

**Justification Of Final Rating:**

The additional explanations provided during the rebuttal have improved my understanding of the paper, which was previously limited due to some structural issues (e.g. the baselines were not clearly introduced). The authors provide convincing evidence that the targeted cyclic approach improves performance. However, to substantiate the claim of fairness, it would have been necessary to include the proposed ablation study.

**Justification Of The Preliminary Rating:**

The idea of targeted synthesis using uncertainty quantification seems promising, however, the aspect of improved fairness solely by adding demographic identity information as text prompt is not sufficiently supported by evidence. Further, the proposed method could be introduced in a more structured way.

**Questions To Address In The Rebuttal:**

- Address the points raised in the detailed comments
- Ablation study for demographic identity text prompts

---

> ### Author Response · Authors · 2026-01-25
>
> We thank the reviewer for the detailed feedback and constructive suggestions.
>
> **Difference to ControlNet.**
>
> ControlNet serves as the conditional diffusion backbone in our generator, but our contribution is not a modification of ControlNet itself. Instead, the novelty lies in the end‑to‑end **cyclic** framework that couples the classifier and the generator through a behavior‑driven Sorter that identifies overconfident errors and triggers targeted synthesis and pruning. This closed‑loop mechanism is absent in vanilla ControlNet.
>
> The ablation in Sec. 5.3.2 directly supports this distinction: when ControlNet is used alone (i.e., without cyclic refinement), its generated images yield consistently lower downstream performance (Table 6). After fine‑tuning ControlNet on the hard‑case subsets selected by our Sorter, the resulting synthetic images become substantially more effective for training, demonstrating that the gains come from our targeted cyclic refinement rather than from ControlNet itself.
>
> Furthermore, Sec. 5.2.2 shows that simply enlarging the synthetic dataset (2Synth/5Synth) does not achieve comparable improvements, confirming that the benefit is not due to generating more samples with ControlNet but due to our behavior‑driven selection of overconfident failure regions. As shown in Table 1, the Female subgroup accounts for 57.72% of the dataset, yet all methods still obtain a lower AUC on Female than on Male (Table 3), indicating that the Female cases are intrinsically more challenging. This highlights why subgroup disparities cannot be resolved by increasing sample counts alone and why targeted synthesis is necessary to improve fairness.
>
>
>
> **Cleaned FairVLMed.**
>
> We agree this term may cause confusion. In this paper, “cleaned FairVLMed” refers to the publicly released curated subset used in our experiments (7,363 images, as provided by Wang et al., 2025). We will revise the wording in the final version to clearly specify the exact data version and preprocessing, ensuring full reproducibility and fair comparison across methods.
>
>
>
> **Missing “images”**
>
> Thank you for spotting this. We will correct the wording in Sec. 4.1 in the final manuscript and add the missing word ‘images’ to the dataset description for clarity.
>
>
>
> **Baselines and Differences from the proposed method**
>
> For clarity, we will add brief introductions of all baselines in the revised manuscript.
> FairCLIP (Luo et al., 2024a) is an optimal‑transport–based method that improves fairness by minimizing the Sinkhorn distance between the global distribution and demographic‑specific distributions in vision‑language models.
> FIN (Luo et al., 2024b) introduces fair identity normalization to enhance intra‑group feature consistency, achieving strong and fair performance in glaucoma diagnosis.
> FairDomain (Tian et al., 2024) proposes a plug‑and‑play fair identity attention module that adjusts feature importance using demographic attributes, improving fairness under domain shift.
> FairVision (Luo et al., 2023) introduces the Fair Identity Scaling, which assigns loss weights using both individual‑level and group‑level scaling to improve fairness across race, gender, and ethnicity.
>
> Importantly, none of these baselines perform behavior‑driven hard‑case selection, cyclic refinement, or shape‑controlled generative augmentation. These components are unique to our proposed CCG framework and are responsible for the observed improvements in both performance and fairness.
>
> To ensure fairness, all methods are trained with exactly the same number of CNN fine‑tuning epochs as in our full CCG pipeline, and we will make this protocol explicit in the final version.
>
>
>
> **Ablation study for demographic identity text prompts.**
>
> We appreciate the reviewer’s suggestion. We agree that an ablation removing demographic identity prompts would strengthen the empirical evidence. We will include this ablation in the final version of the manuscript and report the corresponding fairness and performance metrics to clarify the role of demographic prompts within CCG.

---

> > ### Comment · Reviewer_8siR · 2026-01-29
> >
> > Thank you for the extended explanation. Especially the introduction to the baselines helped my understanding of the paper. I would be interested in seeing the outcome of the ablation study for demographic identity text prompts. Good luck with your submission!

---

> > > ### Author Response · Authors · 2026-02-01
> > >
> > > We thank the reviewer for the positive feedback and for the interest in the ablation study. It is important to note that the demographic identity text prompts serve as prerequisite pseudo‑labels for assigning population attributes to the generated images. Removing them would eliminate the attribute “ground truth” required for downstream fairness evaluation, which limits our ability to conduct such an ablation within the current framework. We will include a discussion of this limitation in the revised manuscript.

---

### Official Review · Reviewer_gRtb · 2026-01-09

**Confidence:** 5
**Preliminary Rating:** 3
**Final Rating:** 4

**Summary:**

This work developed a framework called Cycle Control Generation to address demographic bias and data scarcity in AI-based glaucoma diagnosis. It combines both a classification model and a guided stable diffusion model to produce synthetic medical images that are balanced across demographics. Within this framework is a mechanism that uses uncertainty to identify overconfident samples and uses them for data augmentation. Evaluated on the Harvard-FairVLMed dataset, the framework demonstrates that it does improve diagnostic accuracy

**Strengths:**

The proposed framework does exhibit several interesting directions and values.
1. Strategy for data augmentation: By identifying and generating synthetic variants of samples where the model is 'confidently wrong', the framework produces some supervision for the classifier. This is a nice use of uncertainty scores, not only for quality control but also for data augmentation. "Uncertainty-driven data augmentation"
2. Dynamic and Iterative Refinement: The cyclic nature allows for a dynamic update process where the sorter module continuously re-evaluates which samples are challenging as the classifier improves. This makes for a good follow-up to the above point. I guess this is to ensure the model does not overfit to a static set of synthetic images?
3. Ablation experiments: Nice to see the different components that made up the framework evaluated.

**Weaknesses:**

This is a method-driven paper and introduces some complex framework that involves generative models (Diffusion model) and also an iterative training. Some weaknesses I can see. This can also be clarified in the rebuuttal as indication in the questions.
1. The aggressive pruning. THere is no rationale for n=50%
2. The definition of lower quality. When is a synthetic image bad due to generative artifacts and one that is juust a hard case for the classifier?
3. The text and shape conditioning is not described well and sufficiently. Dodes the generative model really made use of these different representations?
4. Expert validation. Honestly not really sure whats going on here. Maybe a clarification.
5. Compute overhead due to the use of diffusion model and also the iterative training.

**Detailed Comments:**

A clearer clarification of the cyclic process. Maybe not an extensive experiment, but there is no evaluation of the Text and shape contribution.

**Justification Of Final Rating:**

The authors clarified the concerns I have over the weaknesses of the paper during the rebuttal. The authors also provided more clarification on the contributions of text and shape, hence the final rating.

**Justification Of The Preliminary Rating:**

Despite the strengths of the paper, there is still so much clarity needed to improve.  The weaknesses outweigh the strengths. Also definitely open to discussing the questions raised and improving the rating.

**Questions To Address In The Rebuttal:**

1. What's the contribution of the text and shape? "This helps generated images better represent diverse subgroups and supports fairness" No results to verify this statement.
2. What's the cycle duration? Implementation details state 5 Cycles, yet it also mentions the CNN classifier is fine-tuned for 30 epochs and the ControlNet for 2 epochs. Whats the relationship between the epochs and cycles?
3. Pruning threshold. Isn't the threshold aggressive at n=50? (Section 4.3)
4. What does lower quality mean here (section 3.2, last paragraph)? If the goal of the framework is to augment the dataset with challenging samples to improve the model. In a way, this is paradoxical.

---

> ### Author Response · Authors · 2026-01-25
>
> Thank you for the careful review and constructive comments. Below we respond to the principal concerns and note planned clarifications for the final manuscript.
>
>
>
> **Text/Shape contribution.**
> Our method conditions synthesis on both demographic/clinical text prompts and optic-disc segmentation masks.
> Text Prompts: These are the sole mechanism for generating subgroup-specific images (e.g., “Female, Asian, with Glaucoma”). Without them, targeted augmentation for fairness would be impossible. Their efficacy is directly validated by expert review: ophthalmologists’ diagnostic judgments on synthetic images agreed ~90% with the generation labels (Sec.5.3.1, Fig.2), proving the text conditions yield clinically label-faithful images.
> Shape Masks: These enforce anatomical plausibility. Experts noted our synthetic images had “clearer optic disc boundaries” and “fewer artifacts.” Quantitative metrics (FID, SSIM in Table 6) independently confirm our method generates higher-quality, more realistic images than the ControlNet baseline.
> Together, they produce the diverse, realistic SLOs that lead to the balanced performance improvements across all demographic groups shown in Table 3.
>
>
>
> **Cycle vs. epoch clarification.**
>
> We apologize for the lack of clarity. A *cycle* corresponds to one iteration of:
> (1) fine-tuning the CNN classifier on $D_s \cup D_g$ for $30$ epochs,
> (2) scoring all samples using the overconfident error $v$,
> (3) selecting the top $m\%$ hard cases from $D_s$ and fine-tuning ControlNet for $2$ epochs, followed by synthetic image generation,
> and (4) updating $D_g$ by adding the newly generated samples and pruning low-quality ones.
>
> This matches the dynamic update process illustrated in Fig.1. In our experiments, we run $5$ such cycles.
> We will revise Sec.4.3 to explicitly clarify the relationship between cycles and the training epochs used for the CNN and ControlNet.
> To make sure that all is fair, all methods employ the same number of CNN fine-tuning epochs as our whole CCG pipeline. We will make this evaluation process clear in our final version.
>
>
>
> **Pruning threshold (n=50%)**
>
> We selected the pruning threshold $n=50\%$ by cross‑validation to balance removal of low‑quality, overconfidently mislabelled synthetic samples (identified by large $v$) against retaining sufficient synthetic diversity to benefit classifier training. Table 5 shows that naive synthetic augmentation (CNN+Synth/2Synth/5Synth) underperforms compared to our cyclic filtering, supporting the need to control $D_g$. During validation we compared different thresholds (e.g., 40%, 50%, 60%) and selected $n=50\%$ based on validation AUC and subgroup performance trade‑offs; this threshold consistently yielded the best balance between generation efficiency and performance gain. We will add a brief sensitivity analysis and the cross‑validation procedure to the final manuscript.
>
>
>
> **Low‑quality samples versus challenging cases**
>
> We clarify that “lower‑quality” synthetics are not synonymous with “challenging” cases. We aim to retain challenging but label‑consistent samples while removing synthetics that are likely mislabeled or condition‑mismatched.
> Concretely, we prune generated images that are severely misclassified with high confidence, i.e., large error $e$ and low uncertainty $u$, yielding a high score $v = e\cdot(1-u)$; such samples are unlikely to reflect the intended generation label and can introduce harmful supervision. By contrast, truly difficult but label‑consistent cases typically induce higher uncertainty $u$, producing lower $v$ and thus being less likely to be removed. This design preserves the framework’s goal of augmenting with informative hard examples while avoiding the paradox of retaining confidently incorrect synthetics.
> We will further clarify this point in the final manuscript.
>
>
>
> **Expert validation**
>
> Expert validation details are provided in Section 5.3.1: two experienced ophthalmologists independently and blindly evaluated 100 randomly sampled synthetic SLO images from $D_g$ (experts reviewed images without diagnostic prompts or generation labels; inter‑rater agreement ICC $>0.95$; expert diagnostic concordance with intended generation labels was nearly 90%, see Fig. 2).
> Evaluation focused on (1) whether glaucoma appearance in generated images matches the diagnostic label in the text prompt, and (2) whether generated images exhibit clinically plausible features.

---

> > ### Comment · Reviewer_gRtb · 2026-01-30
> >
> > Thanks for the detailed explanation and for answering the questions. There is some overlap with related work by Ilanchezian et al. (2025) [1], which generates counterfactual retinal images using diffusion models, which are also clinically relevant. This work is not cited in the manuscript.
> >
> > [1] https://journals.plos.org/digitalhealth/article?id=10.1371/journal.pdig.0000853

---

> > > ### Author Response · Authors · 2026-02-01
> > >
> > > We thank the reviewer for highlighting this relevant work. The study by Ilanchezian et al. (2025) is indeed closely related and helpful for positioning our contribution. We will add this citation and discuss its connection to our method in the revised manuscript.

---

### Comment · Area_Chair_atDY · 2026-01-27
**Engage with reviewers & finalize scores**

Dear Reviewers,

please have a thorough look at the responses by the authors. Please acknowledge the responses and engage in discussion if anything remains unclear. Please update your final rating by clicking “Edit” → “Official Review” and providing the Final Rating by February 1st 2026 (23:59 AoE).

Best
AC

---

### Meta-Review · Area_Chair_atDY · 2026-02-03

**Recommendation:** Accept (Poster)
**Confidence:** 3

**Metareview:**

The paper received overall positive reviews, with final ratings of "Strong Accept", "Weak Accept" and "Borderline". The "Borderline" reviewer maintained his/her rating because a suggested ablation study regarding the fairness claims was not introduced during the rebuttal period. As the rebuttal phase was short, these additional experiments may have been beyond the scope. I therefore side with the reviewers arguing to accept the paper.

---

### Decision · Program_Chairs · 2026-02-13

Accept (Poster)